# Vapor-deposited zeolitic imidazolate frameworks as gap-filling ultra-low-k dielectrics

Mikhail Krishtab[1,2], Ivo Stassen[1,2], Timothée Stassin [1,2], Alexander John Cruz [1,2], Oguzhan Orkut Okudur[2,3], Silvia Armini[2], Chris Wilson[2], Stefan De Gendt[2,4] & Rob Ameloot[1]

The performance of modern chips is strongly related to the multi-layer interconnect structure that interfaces the semiconductor layer with the outside world. The resulting demand to continuously reduce the k-value of the dielectric in these interconnects creates multiple integration challenges and encourages the search for novel materials. Here we report a strategy for the integration of metal-organic frameworks (MOFs) as gap-filling low-k dielectrics in advanced on-chip interconnects. The method relies on the selective conversion of purpose-grown or native metal-oxide films on the metal interconnect lines into MOFs by exposure to organic linker vapor. The proposed strategy is validated for thin films of the zeolitic imidazolate frameworks ZIF-8 and ZIF-67, formed in 2-methylimidazole vapor from ALD ZnO and native $CoO_x$, respectively. Both materials show a Young's modulus and dielectric constant comparable to state-of-the-art porous organosilica dielectrics. Moreover, the fast nucleation and volume expansion accompanying the oxide-to-MOF conversion enable uniform growth and gap-filling of narrow trenches, as demonstrated for 45 nm half-pitch fork-fork capacitors.

[1] Department of Microbial and Molecular Systems, Centre for Surface Chemistry and Catalysis, KU Leuven - Celestijnenlaan 200F, 3001 Leuven, Belgium. [2] imec - Kapeldreef 75, 3001 Leuven, Belgium. [3] Department of Materials Engineering, KU Leuven, Kasteelpark Arenberg 44, 3001 Leuven, Belgium. [4] Department of Chemistry, KU Leuven - Celestijnenlaan 200F, 3001 Leuven, Belgium. Correspondence and requests for materials should be addressed to R.A. (email: rob.ameloot@kuleuven.be)

Since the invention of integrated circuits (ICs) 60 years ago, there has been a persistent incentive toward the miniaturization of IC components. An indispensable part of every contemporary IC is a multilevel wiring system fabricated on top of the semiconductor device layer. As transistors get smaller and more densely packed, the complexity and the impact on performance of the on-chip interconnects rises. Modern chips can incorporate several billion transistors and require multilayered interconnects that have wire dimensions below 30 nm in diameter near the silicon surface[1,2]. The nonzero resistance ($R$) and capacitance ($C$) associated with the metal wires and the insulating medium between them induce cross-talk noise, limit the speed of signal propagation, and contribute to the power consumption of a chip[3]. Both $R$ and $C$ tend to increase with further interconnect miniaturization, an effect particularly noticeable at wire diameters and spacings below 50 nm. To address this challenge, the materials used for on-chip interconnects have been significantly diversified in last two decades[4]. Two major changes concerned (i) the replacement of traditional dielectrics such as silicon oxide with "low-k" materials featuring a dielectric constant (or "k-value") <4.3 and (ii) the introduction of low-resistivity metals instead of aluminum. Currently, copper is the metal of choice due to its high conductivity and good resistance to electromigration, while state-of-the-art low-k dielectrics are typically porous organosilica glasses (OSGs) deposited from plasma or solution[5–8]. Due to difficulties associated with direct plasma etching of copper films[9], copper interconnect wires are formed via a "damascene" process[10]. This metallization route relies on first patterning the dielectric layer via anisotropic plasma etching before Cu electroplating and planarization. Plasma-induced damage of new potential low-k materials, especially porous ones, during this patterning step is a major roadblock to achieve k-values < 2.4[11,12]. An alternative metallization approach is to first pattern a sacrificial layer that can be removed after the formation of metal interconnect wires, leaving the latter as a patterned top layer[13]. In such an integration scheme, the low-k dielectric is subsequently deposited and needs to fill the gaps between the metal wires. Driven by the steep increase in copper resistivity at wire dimensions below 50 nm[14], this and other alternative metallization strategies relying on initial formation of metal pattern are quickly gaining attention[15–18], together with a shift to beyond-Cu metals that perform better at these dimensions (e.g., Co and Ru)[19–22]. The common challenge shared between these alternative metallization approaches is gap filling the narrow trenches (width < 50 nm, height/width ratio >2) between metal interconnect lines with the low-k dielectric[23,24]. For the current state-of-the-art OSG dielectrics deposited by plasma-enhanced chemical vapor deposition or spin-on methods, this task is not straightforward because of local deposition nonuniformity and postdeposition material shrinkage upon porogen removal. While current ultra-low-k dielectrics thus seem unsuitable for novel, plasma-damage free integration, the alleviated plasma stability requirement allows to significantly widen the range of potential low-k dielectrics.

One promising class of dielectric materials are metal-organic frameworks (MOFs)[25], which possess many properties sought for in interconnects: (i) uniformly distributed pores with diameters < 2 nm that significantly reduce possible infiltration during subsequent processing[26,27] and makes the porous dielectric applicable in sub-10 nm intermetal spacings[28,29]; (ii) a low dielectric constant; and (iii) a suitable mechanical stability[6,7]. The potential of MOFs as advanced low-k dielectrics was first recognized by Zagorodniy et al., who used a simple Clausius–Mossotti model to estimate the dielectric constants of various MOFs[30]. Eslava et al. provided the first experimental validation of the k-value of solution-deposited films of ZIF-8 ($k = 2.33$)[31], a well-studied thermally and chemically stable MOF consisting of Zn(II) nodes

and 2-methylimidazolate linkers[32]. Since then, other MOFs have been studied as potential low-k dielectrics[33–36]. Despite these encouraging results, no integration of MOFs in interconnects has been shown to date, mainly because typical MOF thin film growth methods are solution based, and therefore hard to scale and mostly incompatible with microelectronics fabrication infrastructure[37]. Nevertheless, recent advances in vapor-phase deposition of MOF films[38–40] bridge this gap and make integration in on-chip interconnects possible.

In this work, we adapted the method of chemical vapor-phase deposition of zeolitic imidazolate frameworks (ZIFs) proposed by Stassen et al. (MOF-CVD)[37] to achieve gap filling of narrow spacings between metal interconnect lines (90 nm pitch, 45 nm trench width, and 90 nm trench height). In MOF-CVD, a metal oxide precursor film is consumed through a vapor-solid reaction with the organic linker and converted to a porous MOF film. In the case of ZIF-8 MOF-CVD, a previously deposited ZnO layer reacts with 2-methylimidazole vapor (2-HmIm). Importantly, ZIF-8 film growth is selective as it is restricted to the area covered by ZnO and the final film is significantly thicker than the initial ZnO layer (~16–17 times, based on theoretical crystal densities), because of the added organic content and the porous nature of the MOF. Based on these characteristics of the MOF-CVD process, we propose two general integration routes for MOFs as gap-filling dielectrics into the interconnect metallization layer (Fig. 1). In both routes, gap filling relies on localized material expansion during the precursor-to-MOF conversion. Route A targets gap filling the trenches between passivated metal lines by first depositing a conformal oxide/hydroxide layer as an MOF precursor. This route can be an optimal choice when a metal diffusion barrier is required (e.g., Cu interconnects). Route B is arguably more elegant since the MOF precursor layer is formed by the controlled oxidation of the metal line surface[41–43]. Since this route results in the direct contact between the metal and the MOF, low metal ion diffusion through the dielectric is needed to avoid electrical reliability issues. In addition, this route requires that the oxide-to-MOF conversion strictly halts at the metal/metal oxide interface so that the interconnect wire remains intact. Route B is particularly interesting for Co metallization since cobalt wires may not need a metal diffusion barrier[44] and at the same time form a $CoO_x$ precursor skin layer through surface oxidation[43]. In this proof-of-concept study, we focus on Zn(II)-based ZIF-8 and its isostructural Co(II)-based analogue ZIF-67 to demonstrate the feasibility of Routes A and B, respectively.

## Results

**Validation of the MOF-CVD process on blanket films.** Figure 2a schematically shows the precursor-to-MOF conversion that was performed in a simple glassware reactor (Supplementary Fig. 1). Starting from 5.8 nm atomic layer deposition (ALD) ZnO on top of an Si substrate and from 3.2 nm native $CoO_x$ on top of 20 nm PVD Co, the conversion resulted in 24.0 nm ZIF-8 and 28.9 nm ZIF-67, respectively, as estimated from scanning electron microscopy (SEM) cross-sectional images (Supplementary Fig. 3b). The crystalline structure and composition of the films were confirmed by GI-XRD (Fig. 2b) and X-ray photoelectron spectroscopy (XPS) (Supplementary Table 1).

The "effective" thickness expansion factors of 4.1 for ZnO and 9.0 for $CoO_x$ based on ellipsometry and SEM data (Fig. 2a) appear significantly lower than the theoretical expansion of 17.0 and 20.4 calculated for wurtzite hexagonal ZnO and cubic CoO, respectively. The theoretical expansion factor calculated based on the metal density of ideal ZnO and CoO crystals is an upper limit since (i) precursor films can contain defects (e.g., hydroxyls) and therefore can have a lower metal content compared with pure

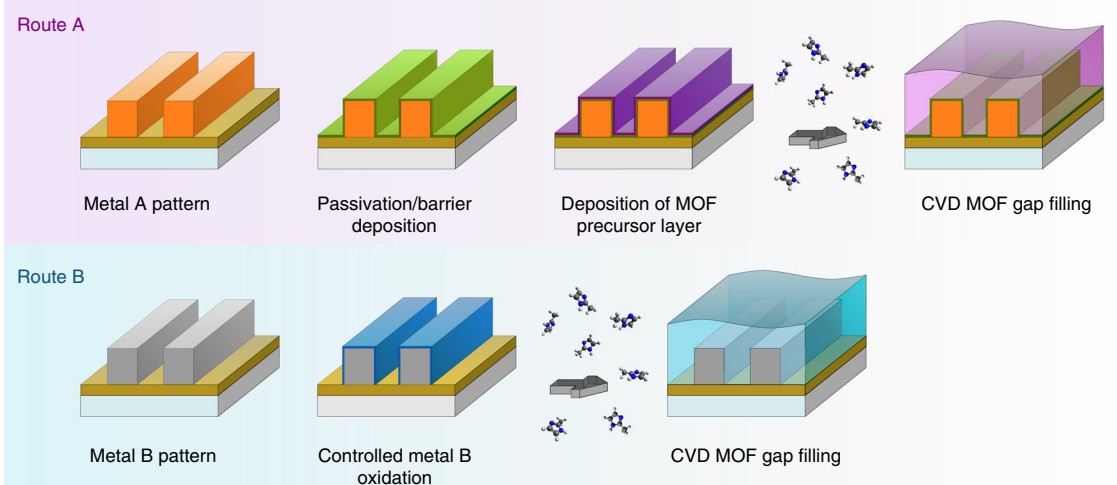

**Fig. 1** Two proposed routes for the integration of ultra-low-k MOF dielectrics in on-chip interconnects via the MOF-CVD process. Routes A and B differ in how the MOF precursor layer is formed around the interconnect wires. In Route A, metal oxide to be converted into MOF is deposited after passivation of metal lines, while Route B relies on selective conversion of metal oxide formed through controlled oxidation of the metal pattern

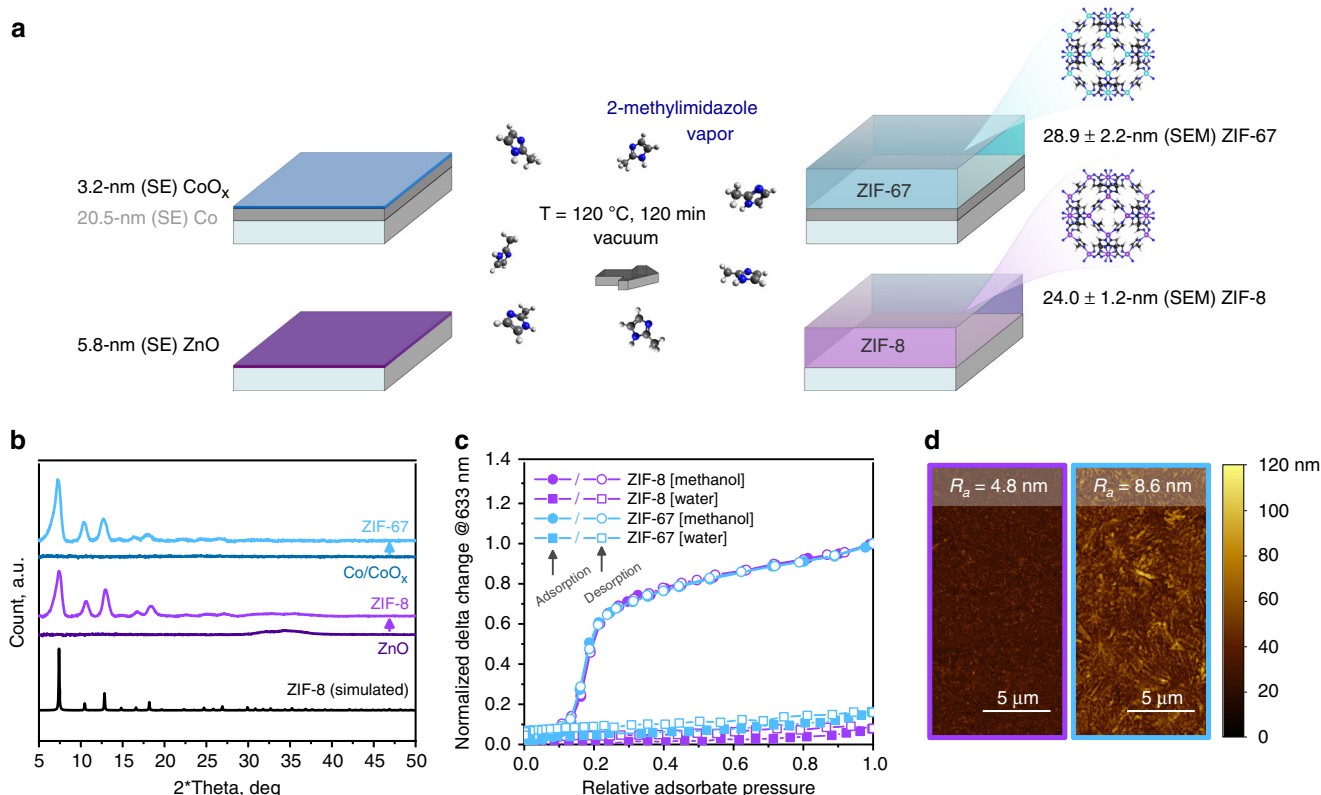

**Fig. 2** Validation of the MOF-CVD process and characterization of the deposited MOF thin films. **a** Schematic representation of the conversion of ALD ZnO and native CoOₓ to ZIF-8 and ZIF-67 and the corresponding increase in thickness as measured by spectroscopic ellipsometry (SE) and from SEM cross-sectional images. **b** Baseline-corrected GI-XRD diffraction patterns together with simulated powder diffractogram for ZIF-8. **c** Ellipsometric porosimetry with methanol and water as adsorbates. The amount of adsorbate corresponds to the change of the ellipsometric angle Delta (@633 nm) relative to the value recorded before introducing probe molecules. The values are normalized against the Delta change measured at methanol saturation pressure. **d** AFM topography images of MOF-CVD films: ZIF-8 (purple frame) and ZIF-67 (light blue frame)

oxide crystals and (ii) the precursor-to-MOF conversion might be incomplete. Both effects were investigated for the ALD ZnO and native $CoO_x$ precursor layers.

For the ALD ZnO film (45 DEZ/$H_2O$ cycles at 120 °C), the volume concentration of Zn atoms was estimated by combining the Zn atom areal density ($30.8 \times 10^{15}$ atoms per cm²) assessed by Rutherford backscattering spectroscopy (RBS) (Supplementary Fig. 4) and the thickness (8.9 nm) measured by ellipsometry. The resulting density of Zn atoms equal to $3.45 \times 10^{22}$ atoms per cm³ is lower than that of wurtzite ZnO by ~18%, which results in the

proportional reduction of the thickness expansion and an expansion factor of 13.9. The lower Zn content of the ALD ZnO film likely originates from pending Zn–OH groups that remain unreacted during the low-temperature ALD process[45]. Based on the corrected expansion factor, it can be calculated that roughly 4 nm ZnO remains at the interface between the ZIF-8 film and the substrate. However, due to its relatively high conductivity[46,47], the remaining thin ALD ZnO layer has negligible impact on the measured MIM capacitance.

The situation is more complicated for the native $CoO_x$ layer formed on top of the sputtered Co film. While there is no doubt about the formation of ZIF-67 (based on GI-XRD, porosimetry and XPS data), the presence of metallic Co prevents the use of the RBS methodology to correct the expansion factor and estimate how much $CoO_x$ remains. Therefore, the aforementioned theoretical expansion factor for perfectly crystalline CoO was used to estimate the maximum thickness of the remaining $CoO_x$ layer as 1.8 nm. In reality, the remaining $CoO_x$ layer will be thinner due to a lower effective expansion factor[48,49]. Because of its higher resistivity, the presence of $CoO_x$ at the interface between metallic Co and ZIF-67 should be taken into account for the estimation of the MOF k-value. However, the impact of this layer was estimated to be negligible (Supplementary Fig. 5). Although in the current experiment some $CoO_x$ remains, the conversion process would in any case be expected to stop at the metallic Co surface, thus preserving the metal line in analogy with a recent study on metallic Zn surfaces[50].

Ellipsometric porosimetry (EP) is a powerful technique for the evaluation of thin porous films. It is based on in-situ ellipsometry monitoring of the refractive index changes in the porous layer as a result of the adsorption/desorption of a probe molecule supplied at well-controlled partial pressures[51–53]. For a qualitative assessment of the resulting isotherms, one can avoid fitting the recorded Psi/Delta values by plotting the relative changes of these optical parameters as a function of adsorbate pressure (e.g., Delta at 633 nm) (Fig. 2c). Such EP data obtained with methanol as a probe molecule show that both ZIF films display identical adsorption/desorption behavior, characterized by a step in the isotherm in the partial pressure range 0.1–0.2. This observation is consistent with previous reports of methanol adsorption measured for ZIF-8 powder samples[54,55]. The absence of secondary sorption at higher relative pressures, characteristic of intercrystallite mesoporosity, positively distinguishes the MOF-CVD ZIF layers from the pure-silica zeolite (PSZ) films that also have been considered as potential low-k dielectrics[56,57]. Another advantage of the ZIF-8 and ZIF-67 frameworks is that they are intrinsically hydrophobic due to the methyl group on the organic ligand pointing toward the pore interior[58]. The MOF-CVD ZIF films do not absorb moisture, even at high relative humidity, as evidenced by water-based EP. While PSZ powders are also intrinsically hydrophobic[59], thin films of PSZ and PSZ-like materials often suffer from moisture uptake and a related k-value increases due to silanol-rich grain boundaries or the presence of amorphous silica[60–62].

### Balance of dielectric constant and Young's modulus (YM) in MOF-CVD ZIF films.
When considering MOFs as low-k dielectrics, it is essential to get an accurate value of the dielectric constant at low frequencies (<1MHz), conventionally used for assessment and comparison of various low-k materials. Measurements on MOF-CVD films offer an advantage compared with the MOF powder pellets commonly studied, not only because of the absence of intercrystal voids[63,64] or potential pelletization-induced damage[65], but as well because such high-quality pinhole-free films (Fig. 2b) allow to employ the standard measurement protocol for low-k dielectric coatings[66]. The k-value of the

MOF-CVD ZIF layers was extracted from capacitance data measured at frequencies in the range 10 kHz—1 MHz on planar metal-insulator-metal (MIM) stacks. The latter are formed by e-beam evaporating round Pt electrodes of different diameter (195, 107, and 62 μm) on top of the MOF-CVD films deposited on conductive substrates (Fig. 3a). Measuring the capacitance on Pt

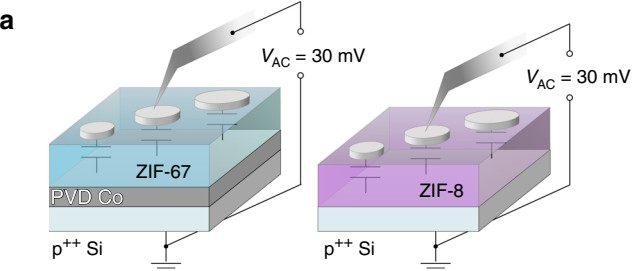

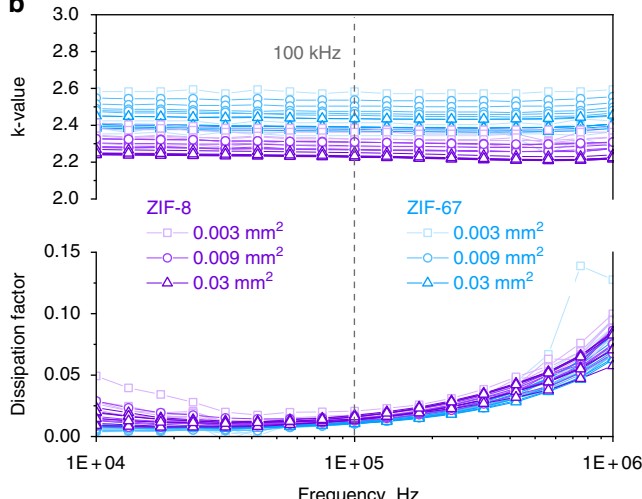

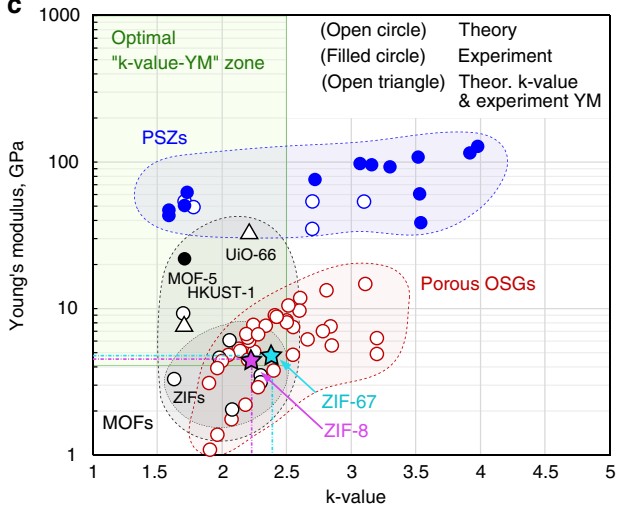

**Fig. 3** Dielectric constant and Young's modulus of MOF-CVD ZIF-8 and ZIF-67 films. **a** experimental setup for k-value measurements. **b** k-value and dissipation factor spectra (10 kHz–1 MHz) measured on MIM capacitors with three different top electrode areas. **c** Positioning of MOFs on the Young's modulus/k-value map in comparison with other classes of advanced low-k dielectrics. The Young's modulus and k-value data measured on MOF-CVD ZIF-8 and ZIF-67 films (this work) are marked with stars. The theoretical and experimental values for the Young's modulus and k-value in the graph are taken from[31,56,63,72,74,86–100]

electrodes of different sizes allows to check the uniformity and continuity of the MOF-CVD layers. Similar to good insulators, such as silicon oxide or OSG low-k dielectrics[66], the $k$-values measured for both ZIF-8 and ZIF-67 films demonstrate nearly no dispersion in the employed frequency range (Fig. 3b). In addition, the $k$-values are insensitive to heating up to at least 150 °C (Supplementary Fig. 6). The absence of any significant conductivity or low-frequency dipole relaxation modes leading to dissipation of electromagnetic field energy is also confirmed by a low dissipation factor ($D = 0.01$–$0.02$ at 100 kHz). The $k$-values extracted from the smaller capacitors tend to be slightly higher due to the effect of fringing capacitance[66,67]. The contribution of the latter is negligible for the $k$-values estimated on the largest electrodes: $2.23 \pm 0.11$ for ZIF-8 and $2.39 \pm 0.18$ for ZIF-67 at 100 kHz. The uncertainty for these $k$-values originates from the statistical variation of the thickness extracted from SEM images (Supplementary Fig. 3b). Interestingly, the obtained $k$-value for MOF-CVD ZIF-8 agrees well with earlier measurement of $2.33 \pm 0.05$ (at 100 kHz) by Eslava et al. on 10–30 times thicker solution-deposited ZIF-8 films[31]. While the $k$-value of Zn-based ZIFs was shown to scale with the framework density[63], the impact of the metal ion node has not yet been reported. According to our results, the Co-based ZIF-67 possesses a slightly higher $k$-value than its Zn-based analog. A similar relation was reported for a pair of isostructural tryptophanate frameworks based on Zn (II) and Co (II) ions[68]. As the dielectric constant strongly depends on the polarizability of the bonds in the framework, the increased $k$-value of Co-based MOFs may be due to the more polar Co-N bonds as compared with Zn-N[69].

Another important characteristic for potential low-k dielectrics is their mechanical stability, typically assessed via nanoindentation (NI) and expressed as the YM. Since conventional data analysis techniques cannot provide a correct interpretation of measurements on sub-100-nm films due to substrate effects, we used an iterative data analysis algorithm proposed by Li and Vlassak[70] to extract the elastic modulus of the ZIF layers. This method was previously shown to yield consistent results for a wide range of thin low-k dielectric films[71]. Besides the low thickness, the measurement was challenged by the polycrystalline texture of the MOF-CVD films, by the associated surface roughness and crystal facet-dependent elastic properties. Nevertheless, the average YM for ZIF-8 MOF-CVD films ($3.44 \pm 1.42$ GPa) is in good agreement with the previously published

isotropic (Voigt–Reuss–Hill averaged) experimental and theoretical YM values of 3.15 GPa[72]. The slightly higher YM of ZIF-67 MOF-CVD films ($3.79 \pm 1.83$ GPa), reported here for the first time, could be ascribed to the larger stiffness of Co-based framework as compared with ZIF-8[73]. The latter was attributed to the more ionic character of Co-N bond.

To position the MOF-CVD ZIFs with respect to other porous materials considered as low-k dielectrics, the YM and $k$-values reported here were included in a summary plot of experimental and theoretical data for MOFs, PSZs, and porous OSGs (Fig. 3c). The YM of state-of-the-art porous OSG dielectrics clearly decreases with a reduction of the $k$-value. This negative effect of the reduced density on the YM can be alleviated by incorporating stronger bonds[74,75], by introducing ordered instead of random pores[29], and by increasing the connectivity of the nodes[76,77]. Although the first two strategies may enable incremental improvements of the $k$-value versus YM trade-off for organosilica materials, breakthroughs are expected only from a transition to a different class of dielectrics. PSZs occupy an attractive area in the top left corner of the plot, thanks to their crystallinity, microporosity, and strong covalent bonding. However, there is no scalable method yet for the deposition of PSZ films that produces uniform hydrophobic polycrystalline coatings without intercrystal porosity and a high concentration of silanol defects. MOFs lie mostly in the ultra-low-k range ($k < 2.5$) and are spread along the YM axis, as expected based on their diversity in terms of bond type and framework topology. Within the MOF family, ZIFs occupy the lower YM range, overlapping with porous OSGs. Although ZIFs are equivalent to the latter state-of-the-art dielectrics in terms of mechanical and dielectric properties, their deposition via MOF-CVD has the distinct advantage of enabling gap filling and therefore future interconnect integration schemes.

**Gap-filling performance of MOF-CVD ZIFs.** The gap-filling test for MOF-CVD ZIF-67 was performed on 45 nm pitch fork–fork capacitors formed by 90 nm thick copper lines passivated with a thin layer of PECVD SiCN diffusion barrier (~3 nm). A conformal film of 2 nm PEALD SiN$_x$ was deposited to improve the nucleation of a nominally 4 nm CVD Co film that was exposed to air to form a CoO$_x$ layer. The result of the precursor-to-MOF conversion process can be seen in Fig. 4a. The bright-field transmission electron microscopy (TEM) cross-sections along with the EDS element maps before and after exposure to 2-HmIm

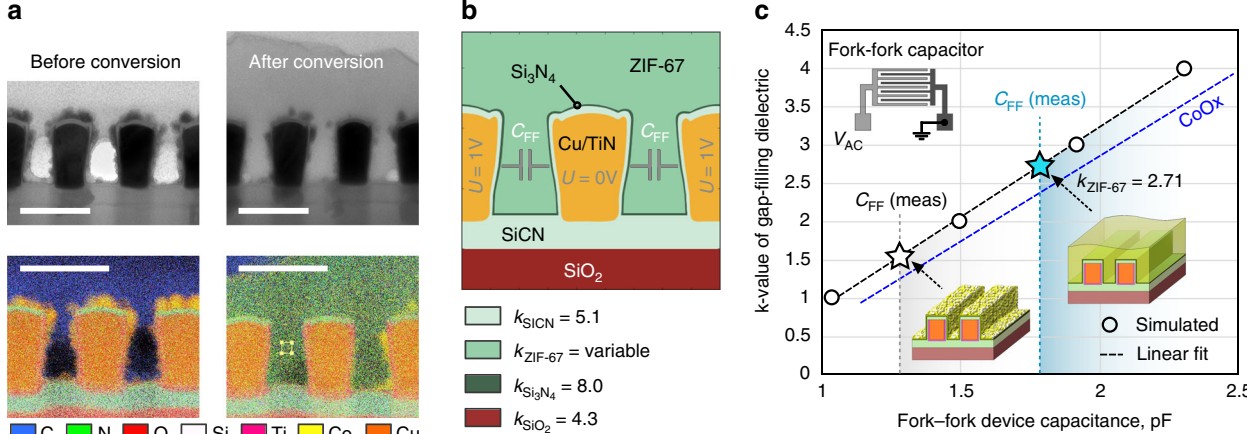

**Fig. 4** Imaging and electrical evaluation of 45 nm half-pitch fork–fork capacitors with oxidized CVD Co layer before and after the conversion reaction. **a** TEM images and EDS elemental maps recorded on lamella cut out from the capacitor device, scalebar is 100 nm. **b** A 2D model of the device cross-section employed for finite-element capacitance ($C_{FF}$) simulation. **c** $k$-value determination of the gap-filling ZIF-67 phase. The black curve represents capacitance simulation data based on the TEM cross-sections in the absence of defects in the gap-filling layer. The blue curve corresponds to similar simulations assuming the worst-case scenario of 20 nm CoO$_x$ particles in the gap-filling phase

vapor provide insight into the structural and compositional changes. The poor nucleation of the CVD Co layer on top of $SiN_x$ results in a rough and non-conformal layer of oxidized Co agglomerates with a size of 10–15 nm on top of the metal lines and 2–3 times smaller inside the trenches. During the vapor-phase conversion nearly all CVD Co is consumed and despite the initial non-uniform distribution of the Co precursor, the space in between and above the passivated copper lines is filled with ZIF-67, as also evidenced by the homogeneous Co and N EDS signal (Supplementary Fig. 7). To demonstrate that the voids in the ZIF-67 phase observed by TEM (Supplementary Fig. 8) are due to the beam damage or the preparation of the 30–50 nm thick lamellas[78,79], additional SEM cross-sectional images were recorded for the same fork–fork capacitors (Supplementary Fig. 9). The cross-section surface was prepared by manually cleaving the substrate and depositing 1 nm of Pt to eliminate any beam-induced damage. No voids can be observed in these high-resolution SEM cross-sections.

To evaluate the gap-filling performance of MOF-CVD ZIF-67, the effective $k$-value was extracted by means of 2D capacitance simulations. In these simulations, the capacitor geometry was taken from the TEM cross-sectional images (Fig. 4b), and the dielectric constant of the gap-filling medium was varied between 1.0 and 4.0 to generate a calibration curve (Fig. 4c). The resulting effective $k$-value of the gap-filling material determined from the capacitance data at 100 kHz (Supplementary Fig. 10) is 2.71 and only slightly exceeds that of the flat ZIF-67 film (Fig. 3b).

$CoO_x$ residues occur largely on top of the passivated Cu lines and are caused by CVD Co agglomerates with dimensions in some cases exceeding 10–15 nm. To estimate the possible impact of these defects on the extracted ZIF-67 $k$-value, additional 2D capacitance simulations were performed (Supplementary Figs 11 and 12). The $k$-value of the $CoO_x$ residues was assumed to be equal to that of CoO ($k = 12.9$)[80]. Since these simulations are two dimensional, defects defined in a cross-section geometry extend in the 3rd dimension, along the Cu-lines. Therefore, the effect of defects on the simulated capacitance is strongly amplified and should be considered a worst-case scenario. Nevertheless, the addition of unconverted $CoO_x$ residues decreases the extracted ZIF-67 $k$-value only by 0.35 for the largest simulated defect dimensions (Fig. 4c). Noteworthy, despite the relatively small effect of these oxide residues on the overall capacitance, their impact on the leakage current between the capacitor forks can be severe (Supplementary Fig. 13). These effects are exclusively caused by the poor nucleation and agglomeration of CVD Co on the surface of the demonstrator capacitors. In the proposed integration Route B (Fig. 1), the use of Co metal lines would eliminate such issues since the $CoO_x$ precursor would conformally cover the lines.

The fork–fork capacitors were also used to check gap filling by MOF-CVD ZIF-8 starting from ~6 nm ALD ZnO deposited conformally on top of the $SiN_x$ passivation layer. The corresponding TEM cross-sections (Supplementary Fig. 14) confirm the successful filling of 45 nm wide trenches with ZIF-8. Even though some unconverted ZnO remains, complete gap filling is evidenced by the homogeneous Zn and N EDS signal.

## Discussion

In summary, MOF-CVD ZIF films do not only demonstrate dielectric and mechanical characteristics competitive with state-of-the-art porous OSG dielectrics but may outperform them in future integration schemes because of the gap-filling nature of the deposition process. Besides validation of the proposed concept, the gap-filling experiments indicate that further control over the properties of the oxide precursors as well as the conversion

process is desirable to avoid precursor residues. To achieve this goal, the MOF-CVD process will have to be implemented as an automated cleanroom unit operation instead of through the use of simple glassware reactors. Efforts are underway to achieve this goal and will be reported as a separate study. When implementing gap-filling MOF-CVD dielectrics, also the further integration steps will have to be optimized. Different process flows can be proposed (Supplementary Fig. 15), in each of which potential points of concern with respect to MOF integration need to be investigated and addressed if necessary: (i) planarization of MOF films when the as-deposited roughness is too high for direct deposition of the next layer; (ii) deposition-induced damage of the MOF layer; (iii) plasma-MOF interactions; and (iv) MOF pore sealing. These topics are barely explored for MOF films and will need more attention in the future.

## Methods

**Preparation of MOF-CVD precursor layers on blanket wafer.** The layers of ALD ZnO and PVD Co were prepared on highly-doped p++ Si substrates. ALD ZnO deposition was realized at 120 °C by 30 cycles of diethyl zinc (DEZ)/water precursor pulses separated by $N_2$ purge steps (Savannah S200, Veeco Instruments Inc.). PVD Co film was sputtered on Ar-plasma precleaned Si substrate (NC7900, Canon Anelva Corp.).

**Preparation of MOF-CVD precursor layer on patterned wafer.** The fork–fork capacitor structures featuring 45 nm line/space width were prepared on p-type 300 mm Si-wafers according to a modified integration route (Supplementary Fig. 2) based on using sacrificial amorphous carbon (a-C) layer to form a pattern of passivated copper wires[13]. The initial stack of layers above the substrate consisted of 1000 nm $SiO_x$, 30 nm SiCN diffusion barrier, 90 nm a-C, and a multilayer hard-mask stack. After formation of a device pattern in the top positive resist coating with 193 nm immersion lithography, the pattern features were then transferred into the underlying a-C film. Following the wet removal of hard-mask residues, the exposed surfaces of a-C/SiCN were coated with 3 nm ALD TiN. The subsequent metallization steps included sputtering of 20 nm Cu seed, electroplating of 500 nm Cu, and chemical mechanical polishing down to the a-C film. The removal of a-C sacrificial layer was done in $He/H_2$ remote plasma. Afterward, the metallic lines were passivated with a non-conformal 3 nm PECVD SiCN barrier layer and then additionally covered with a conformal 2 nm PEALD $SiN_x$ film. The deposition of CVD Co was realized at 200 °C on VECTOR Excel tool cluster (Lam Research Corp.). Before deposition of CVD Co on the $SiCN/SiN_x$-passivated Cu pattern, the growth conditions were optimized on blanket $SiN_x$ surface to obtain $4.0 \pm 1.0$ nm Co layer across 300 mm wafer (assessed by RBS). ALD ZnO deposition on the metal lines passivated with $SiN_x$ layer was performed by applying the same growth conditions as used on blanket wafers (see above).

**Vapor-phase conversion process (MOF-CVD).** For the conversion to appropriate ZIF layer, samples with precursor layers were placed in a glassware reactor (Supplementary Fig. 1). The glassware reactor was connected to a vacuum pump via a manual valve. Upon assembly the reactor was checked for leaks. The glass tube containing 2-methylimidazole powder (99%, CAS #693-98-1, Sigma-Aldrich) was connected to one of the ports of the glassware reactor via another manual valve. The whole setup was placed in a furnace preheated at 120 °C. After the temperature stabilization (15 min), the valve to the vacuum pump was opened, and the reactor was evacuated until pressure stabilization below 10 mbar. The vacuum valve was then closed and the valve to the 2-methylimidazole tube opened. The exposure of samples to vapors of 2-methylimidazole was set to 120 min, after which the precursor valve was closed, and the sample area of the reactor was kept under dynamic vacuum for 15 min to remove the unreacted organic linker from the sample surface and pores of formed ZIF films (activation). Finally, the reactor was let to cool down before the samples could be taken out for further characterization.

**X-ray diffraction (XRD).** XRD patterns were obtained with PANalytical Empyrean diffractometer equipped with Cu $K_\alpha$ X-ray source and Pixcel3D detector. The surface-sensitive grazing incidence XRD mode (GI-XRD) was employed with Omega fixed at 0.2° and 2 × Theta scanned in the range 5–50° with a step size of 0.05°. The diffractograms were then manually baseline corrected using b-spline interpolation.

**Ellipsometric porosimetry (EP).** EP measurements were performed on a custom-built tool featuring a vacuum chamber, spectroscopic ellipsometer (Sentech SE801, 350–850 nm wavelength) mounted on it, and a programmable adsorbate vapor supply platform. The isotherms were recorded at 23 °C (temperature of the vacuum chamber). The temperature of the liquid sources (DIW and methanol) was maintained at 25 °C. The equilibration time at each pressure point was set to 30 s.

**Atomic force microscopy (AFM)**. AFM topography images ($10 \times 20$ μm², $512 \times 1024$ pixels) were recorded in tapping mode with Dimension Edge microscope (Bruker). The image corrections (rows alignment, horizontal scars elimination) and plotting was performed with Gwyddion 2.44 software[81]. The latter was also used to calculate the arithmetic ($R_a$) and root-mean-squared ($R_q$) surface roughness.

**Scanning electron microscopy (SEM)**. SEM images were collected on manually cleaved samples with Hitachi SU8000 microscope operating at 10 kV with secondary electron detector. Besides 70 nm Pt e-beam evaporated on top of ZIF layers (Pfeiffer PLS580) before cleaving, the cross-section surface was additionally sputter coated with 1 nm Pt (Cressington 108) before the inspection. The extraction of average ZIF layer thickness ($d$) from the obtained cross-sectional images was done with a custom Python script based on OpenCV (Open Source Computer Vision Library). This involved locating more than 60 uniformly spaced points at ZIF/Pt interface ($N$). The uncertainty of the average thickness value was expressed via 95% confidence interval $[d - 1.96 \times R_q/\mathrm{sqrt}(N); d + 1.96 \times R_q/\mathrm{sqrt}(N)]$, where $R_q$, root-mean-squared surface roughness calculated from the appropriate AFM images, was used as an accurate estimate of standard deviation of ZIF layer thickness. As a result, the thickness of ZIF-8 and ZIF-67 films used in this work was defined as $24.0 \pm 1.2$ nm and $28.9 \pm 2.2$ nm, respectively.

**X-ray photoelectron spectroscopy (XPS)**. XPS measurements were carried out in angle-resolved mode using a Theta300 system (Thermo Fisher Scientific). Sixteen spectra were recorded at exit angles between 22° and 78° as measured from the normal of the sample. The measurements were performed using a mono-chromatized Al $K_\alpha$ X-ray source (1486.6 eV) and a spot size of 400 μm. Standard sensitivity factors were used to convert peak areas to atomic concentrations.

**Rutherford backscattering spectroscopy (RBS)**. The RBS experiments were performed in a random rotation mode using 1.5 MeV He+ ions from the 6SDH tandem accelerator (National Electrostatics Corporation) equipped with an Alpha-tross ion source. The experimental end-station was a 5-axis goniometer developed at the Forschungszentrum Jülich[82]. The angle between the sample normal and the beam was set to 11°. The solid angle of the PIPS detector was 0.42 msr, the scattering angle was 170°. The beam spot was confined to $1 \times 1$ mm². In-house developed analysis software was used to fit the spectra and to deduce the areal density.

**Nanoindentation (NI)**. NI measurement was performed using Hysitron TI 950 Triboindenter (Bruker). Continuous stiffness measurements (CSM) with a cube corner indenter were employed. The Li and Vlassak method[70] implemented in a custom MATLAB script was applied to extract the YM of the top ZIF layers. Poisson's ratio of the films was assumed 0.44 based on a previous study[72]. At least ten CSM curves were analyzed per sample and the results were averaged. It should be noted that the method is in principle applicable only for bilayer stacks, i.e., thin film on a substrate. This complicates the extraction of elastic properties in case of ZIF-67 film, which is formed on top of 20 nm Co/Si substrate stack. Nonetheless, the additional elastic finite-element simulations demonstrated negligible influence of the intermediate metallic Co layer on the load-displacement characteristics. This can be attributed to the same order of magnitude of YM for Co (200 GPa)[83] and Si (130 GPa)[84] in contrast to much lower value of YM of ZIF films under test (<5GPa). As a result, the presence of Co layer was ignored during the analysis of the stack containing ZIF-67 film.

**Transmission electron microscopy (TEM)**. TEM analysis of the fork–fork capacitor cross-section was done on 30-50 nm thick lamellas prepared with dual beam FIB/SEM Helios NanoLab 450HP (FEI). Before the FIB lift out, the surface of the samples under test was coated with a protective coating consisting of three sequentially deposited layers: a drop-casted organic layer (SoC) soft-baked at 150 °C, e-beam (5 kV) and ion beam (30 kV) deposited Pt layers. As a last step of the lamella preparation, there was a 5 kV ion beam milling step to reduce the surface damage. The TEM images and EDS element maps were captured with Titan³ G2 60–300 TEM (FEI) operating at 60 kV.

**Impedance measurements**. Impedance measurements were performed using high precision LCR-meter E4980A (Keysight Technologies) in the frequency ($f$) range 10 kHz—1 MHz. The measured impedance values were interpreted with an equivalent scheme consisting of serially connected capacitor ($C_s$) and resistor ($R_s$). The dissipation factor ($D$) was estimated as $2\pi \times f \times R_s$. In the case of blanket ZIF films, their $k$-value was calculated from capacitance $C_s$ measured on MIM stacks using the formula for parallel plate capacitor: $k = C_s \times d/(\varepsilon_0 \times S)$, where $\varepsilon_0$ denotes vacuum permittivity, $S$ denotes area of top 70 nm thick Pt circular electrodes e-beam evaporated through a Ni shadow mask (Pfeiffer PLS580), and $d$ denotes average thickness of ZIF layer extracted from cross-sectional SEM images.

**2D capacitance simulation**. The extraction of $k$-value for the gap-filling phase in the case of fork–fork capacitors was performed by adapting the procedure that has been already successfully employed for evaluation of integrated $k$-values of various porous low-k dielectrics[85]. At first, a model of the electrically characterized fork–fork

capacitor is reconstructed based on the previous knowledge about low-frequency dielectric constants of the constituting layers and on their geometrical profiles extracted from TEM cross-sectional images. The extraction of the profiles and layer dimensions of interest was implemented with a help of a custom Python script embedding OpenCV (Open Source Computer Vision Library) functionality. The profiles of the conducting lines (Cu + TiN liner) and of the passivating SiCN and $SiN_x$ films were averaged over six different wire cross-sections. The other dielectric layers in the model were represented by perfect rectangular patches with an average height determined from TEM images. The capacitance simulation was done with a 2D field solver implemented in Raphael program suite (Synopsys). The width of the simulation window could be reduced to only two structure pitches (180 nm) due to employed reflective (Neumann) boundary conditions. The capacitance of a model fork–fork capacitor was calculated as a product of simulated 2D capacitance and the actual capacitor's length (2 cm).

## Data availability

The authors declare that the main data supporting the findings of this study are available within the article and its Supplementary Information file. Extra data are available from the corresponding author upon request.

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

## Acknowledgements

R.A. acknowledges the funding from the European Research Council (No. 716472, acronym: VAPORE) and the Research Foundation Flanders (FWO) for funding in the research projects G083016N and 1501618N and the infrastructure project G0H0716N. I.S. acknowledges the Research Foundation Flanders (FWO) for a postdoctoral fellowship. T.S. acknowledges the Research Foundation Flanders (FWO) for a strategic basic research PhD fellowship. M.K., S.A., and C.W. acknowledge the IMEC's Industrial Affiliation Program on Nano-Interconnects and IMEC's p-line for support in manufacturing of the patterned test wafers.

## Author contributions

M.K., R.A., and S.D.G. designed and supervised the project. M.K. wrote the manuscript with input from all co-authors. The preparation of samples was performed by M.K., I.S., T.S., and A.J.C. and facilitated by C.W. and S.A. GI-XRD measurements were carried out by A.J.C. Nanoindentation data collection and analysis was done by O.O.O. M.K. performed EP and AFM measurements, electrical characterization of planar and fork–fork capacitors. The TEM and XPS analysis data were provided by C.W. The 2D capacitance simulations were set up and implemented by M.K.

## Additional information

**Competing interests:** The authors declare no competing interests.

