## [Peer Review File · Nature Communications]

Reviewers' comments:

Reviewer #1 (Remarks to the Author):

In this paper, the authors describe the integration of metal-organic frameworks (MOFs) as gap-filling low-k dielectrics in one-chip interconnects via a MOF-CVD process. The concept for using ZIF-8 as a low dielectric material was first presented by Eslava et al. in 2010. Since then, some significant studies on the low-k behavior of MOFs and their applications have been reported. e.g. *Coord. Chem. Rev.* 360, 71 (2018); *Nat. Commun.* 7, 11830 (2016); *RSC Adv.* 5, 45213 (2015) and *J. Mater. Chem. C* 2, 3762 (2014). With a fundamental study on dielectric properties and advancement in thin film growth techniques, MOFs were first proposed as future interlayer dielectric materials in 2015. e.g. *ChemElectroChem* 2, 786 (2015). The author is advised to cite all of the relevant literature as mentioned above in the "Introduction".

The authors measured the dielectric constants of two compounds ZIF-8 and ZIF-67. ZIF-8 has already been reported as a low k material, in addition, the CVD technique for ZIFs is also known (refs 31 and 33). In this regard, the novelty is less when it comes to synthesis part of the study. The integration of MOFs in interconnects has not yet been clearly demonstrated. This study is, however, still a nice piece of work with respect to practical applications of MOFs in microelectronics.

Authors should provide data on dielectric constant vs temperature to permit us to understand the polarizability effect of Co-N bonds and Zn-N bonds, which generally increase with increasing temperature and is accompanied by an increase in dielectric behavior.

In addition, on page 6, line 150 and on page 7, line 151 & 158-160, the authors stated that some portion of the unreacted metal-oxide layer could still be present underneath the MOF film at the top of the interconnects. This might result in the tunneling of electrical current across the interconnects through metal-oxide chains and hence, would result in a device with a high leakage current. Leakage current is a significant parameter that needs to be considered in the downsizing of components in an integrated circuit. The value of the leakage current can also provide information concerning the efficiency of these vapor-deposited zeolitic imidazolate frameworks as gap-filling ultralow-k dielectrics. The author should provide information regarding the magnitude of the current leakage across the interconnects.

Reviewer #2 (Remarks to the Author):

The authors present a novel strategy to integrate MOFs on on-chip interconnects using CVD, never used before for MOF low-ks. Moreover, they also present experimental confirmation of the low k value of ZIF-67 films (without intercrystal porosity affecting values). The technological advancement is remarkable. The scientific community will be pleased to read this article and it will trigger novel multidisciplinary research between the MOFs and Microelectronics fields. It is also very well written and appealing for the general audience of Nature Comm. Therefore, I believe it merits publication after the following comments are addressed.

Comments:

Page 3: The sentence "Despite these encouraging results, no integration of MOFs in interconnects has been shown to date, mainly because typical MOF thin film growth methods are solution-based, and therefore hard to scale and mostly incompatible with microelectronics fabrication infrastructure" is arguable. Many spin-on low-k dielectrics have reached high levels of integration despite their solution processing. Novel fields such as halide perovskite solar cells are also based on solution processing. The difficulties/challenges to integrate MOFs as low-ks lie in their thermal and especially chemical stability. ZIF-8 and others are chemically stable in ambient conditions but

the manufacturing of on-chip interconnects is quite demanding. Low-k dielectrics have to withstand harsh processing conditions in different media. See for example the following paper: <https://doi.org/10.1116/1.5038617> where they subject a low-k to H₂ plasma and HF wet cleaning. The sooner this is acknowledged in the scientific literature/community the earlier we will find a solution with novel more chemically-resistant MOFs.

After the CVD MOF grows between the Cu trenches, the films are in excess and kind of wavy. They will need to be planarized and cleaned for the formation of multilevel interconnects of more Cu trenches and insulator. How the authors envisage to continue the multilevel formation with such MOFs that are chemically and thermally not the most stable materials? A bit of discussion on this in the paper, acknowledging the challenges will ensure more rapid progress in the field and more citations.

Page 7: sentence "Another advantage of the ZIF-8 and ZIF-67 films in comparison with pure silica zeolites is that they are intrinsically hydrophobic due to the methyl group on the organic ligand pointing towards the pore interior" is also arguable. Pure-silica zeolites are also intrinsically hydrophobic. It is either their composites with amorphous silica or poorly crystallized films that are not hydrophobic because of the hydroxyls presence in the amorphous part or on defects. Please rephrase the sentence to avoid confusion to readers. So far, highly crystalline pure-silica zeolite films despite being hydrophobic don't have enough porosity for a low k value. There are zeolite types with very high porosity that could show very low k value and excellent hydrophobicity, but their syntheses are too complicated, especially for films and with pure silica content. Please also correct "pure silica zeolite" for "pure-silica zeolite". "Pure" refers to "silica", not to zeolites, and the papers the authors are citing are not strictly pure zeolites.

Some typos such as 2-methylimidazole

Please correct "cross-section SEM images" for "cross-sectional SEM images"

Fig. 2b. If there are no numbers on y-axis, "a.u." is not needed. "Intensity" would be more appropriate than "Count."

Fig. 2c. Please indicate which symbols are for adsorption and desorption

Kind regards,
Dr Salvador Eslava
University of Bath

Reviewer #3 (Remarks to the Author):

See below for the review.

Review of the MS NCOMMS-19-00660:

Comments:

The authors demonstrate a new strategy for the integration of MOFs as gap-filling low-k dielectrics. To use the expanding nature of the transformation of the metal oxide towards a MOF as a gap filling low-k material is quite interesting. However, the approach adopted cannot be stated **novel** as the process is not new for ZIF-8 but only for ZIF-67 and the deposition method is not "classical" CVD but more a chemical infiltration/conversion.

While the manuscript (MS) makes a good impression at the beginning and while the the approach is certainly nice and qualify for publication in a good IF journal, there are several open weak points especially taking into consideration how the authors try to sell their approach. For the level of publication in Nature Communications, the data presented is not strong enough to be published in this journal. There is certainly a lack of persuasive power.

The reasons for not recommending publication in Nature Communications are listed below.

1) As the authors state themselves, the concept is not new but adopted for a proof of principle application (p 4 line 93ff)

2) p. 4 l.100ff MOF expansion by theory 19 - 17 times of the thickness of the oxide... based on theoretical crystal density estimation/calculation, but

2.1) how accurately are/is this model / estimation generally: How is the formation of defects and voids considered or taken into account? The XPS data (the only compositional proof) show imperfect stoichiometries. It may be the case that the model is not relevant for the rest of the MS, but after having a look at page 5 and the big discrepancies that they cannot properly explain, from the reviewers perspective this becomes odd.

2.2) 5.8 nm ZnO --- 24 nm ZIF-8 and 3.2 nm CoOx ---- 28.9 nm ZIF-67

3) Referring to page 5 and beginning of page 6: The model is obviously inaccurate - so how can the authors sell their approach as big breakthrough regarding the problems/issues connected to the prior described state of the art approach? They allegedly! (see later points) solve the problem with the void filling, but on page 6 they demonstrate themselves how difficult precise working with their method is: The expansion of the MOF is highly dependent on the nature of the ALD or native oxide layer and full conversion seemingly difficult to reach. At this state this method is in its infancy and certainly not suitable to be integrated in up-scaled device processing!

4) Also on p.6 they refer to results in another manuscript which they do not provide! As a reviewer we cannot check that - This not good practice and bold. They shall show it.

5) How accurate is ellipsometry for accessing the thickness of MOFs. Do MOFs absorb light of the ellipsometer spectrum? This is not clear... the authors made some thickness measurements for thicker films via TEM and SEM, but not for their devices they describe on page 7ff.... This reviewer is not too convinced when they write about the dielectric constants (k values) and don't explicitly say how thick the MOF films are, by the way. Maybe this reviewer overlooked it, but there seem to be no data on the film thickness for the MIM capacitors and the reviewer would like to know! as thickness influences the k value.

6) p. 7 lines 159-161 Why is it expected? Could the authors comment on it?

7) p. 12 lines 292 - 294: Why does a higher k value imply that ZIF-67 layers are void free before TEM? And how should an electron beam create voids / holes randomly in a MOF layer.... It rather looks like imperfect growth especially in the SEM and TEM of the fork fork structures in the SI!!!! The concept works certainly, but not perfectly and they sell it as if it would – This is not acceptable.

8) p. 13 lines 295-298: This reviewer doesnt understand how the presence of rests of a conductive metal can increase the k of the insulating layer. They should explain their suspicion.

9) Why are the ZIF-Layers in the SI figure 3 colorized? It looks as if they want to hide structural features! They shall decolorize it and just put colored lines on the interfaces between the layers.

10) General question to give an outline to potential readers: How time-and cost intensive is the after-processing after the conversion step? As can be seen from the images of converted MOF on the fork structures, the MOF layer is highly inhomogenously grown.

11) How easily can the thickness be equalized and with which methods (etching will certainly not work)

12) How well suited are MOF layers as substrates for another gas phase deposition or other device fabrication related process. If the topology is like the '*7 Blue Mountains with 7 dwarfs*' then it will certainly be a problem!

13) Line 109: "Controlled oxidation of the metal surface" What are the specific parameters of the controlled oxidation? (Defined oxygen-containing atmosphere? Temperature?)

14) Line 142-148: The expanding factor is calculated based on bulk-crystalline materials "wurtzite hexagonal ZnO" and "cubic CoO", showing higher expansion values than the measured ones. Besides the mentioned explanations: How can the authors be sure that the calculated values are comparable, if the underlying oxide layers may be of amorphous nature? GI-XRD of CoO shows no reflexes and ZnO shows only very small and broad reflexes.

15) Figure 2: Mention that the given reference XRD pattern of ZIF-8 is almost the same as that of ZIF-67, as no reference pattern for ZIF-67 was shown.

16) Line 148: "Non-ideality of the ALD ZnO film". What does non-ideality mean in this case?

17) Line 158-161: "[...], a very thin CoOx layer (< 3 nm) could be still present underneath the MOF film. In both cases, the presence of thin interfacial oxide layers is expected to have negligible impact on the evaluation of dielectric and mechanical properties of the ZIF films, therefore we further treat them as parts of the corresponding MOF layers." If the presence of an interfacial oxide layer is supposed to have a negligible impact on dielectric properties, the authors are asked to cite a relevant paper that supports this idea. Even very thin interfacial oxides have an influence on the k-value determined in MIM-capacitor stacks. This was also the point made above regarding the thickness (point 5).

18) Several references for the Figures are missing in the text (for example Figure 3b). This is sloppy work, especially submitting to high IF journals.

Recommendation:

Although the concept is interesting, the novelty of the work is not suitable to be published in Nature Communications.. The imprecise interpretation as well unacceptable assumptions is certainly an issue that can't be ignored. The points listed above are enough to illustrate that this work presented is not of the quality to be published in Nature Communications. This reviewer suggests revising the manuscript thoroughly giving additional information (as suggested above) and submitting it to a materials chemistry journal.

Feedback to reviewers

Manuscript NCOMMS-19-00660:

Vapor-deposited zeolitic imidazolate frameworks as gap-filling ultra-low-k dielectrics via selective conversion of metal oxide films

Reviewer #1:

1.1 Temperature dependence of dielectric constant (k-value)

Reviewer #1: Authors should provide data on dielectric constant vs temperature to permit us to understand the polarizability effect of Co-N bonds and Zn-N bonds, which generally increase with increasing temperature and is accompanied by an increase in dielectric behavior.

Authors' response:

We would like to thank the reviewer for the valuable remark. A temperature-dependent k-value is not desirable in interconnects, as it would make the performance of interconnects shift with the operating temperature (below 70-125°C for most of silicon-based integrated circuits). Studies on the dielectric properties of MOF films and powders often show a temperature dependence (sometimes irreversible) caused by dielectric relaxations within the framework or by desorption of guest molecules. Nevertheless, the ZIF-8 film in the study by S. Eslava *et al.* showed very little change with heating¹. We observed similar negligible temperature dependency of the k-value of both MOF-CVD ZIF-8 and ZIF-67 films (Supplementary Figure 6).

The temperature dependence of the k-value was collected on the same planar capacitor devices as used for the room temperature measurements. The capacitors with the largest area Pt electrodes (0.03 mm²) were used. During the test, the substrates were placed on a chuck heated in air.

Supplementary Figure 6. Temperature dependence of ZIF-8 and ZIF-67 k-value measured on parallel-plate metal-insulator-metal capacitors with 0.03 mm² Pt top

electrodes. (a) Schematic description of the experiment; (b) k-values of ZIF films extracted from capacitance recorded on the chuck-heated devices in air.

1.2 Leakage current through CVD-ZIF films

Reviewer #1: *In addition, on page 6, line 150 and on page 7, line 151 & 158-160, the authors stated that some portion of the unreacted metal-oxide layer could still be present underneath the MOF film at the top of the interconnects. This might result in the tunneling of electrical current across the interconnects through metal-oxide chains and hence, would result in a device with a high leakage current. Leakage current is a significant parameter that needs to be considered in the downsizing of components in an integrated circuit. The value of the leakage current can also provide information concerning the efficiency of these vapor-deposited zeolitic imidazolate frameworks as gap-filling ultralow-k dielectrics. The author should provide information regarding the magnitude of the current leakage across the interconnects.*

Authors' response:

We agree with the reviewer that leakage current is one of the key parameters for interconnects. However, as also mentioned by the reviewer, the leakage current in such a complex system as on-chip interconnects will depend not only on the intrinsic properties of the dielectric, but also on the defectivity of these materials and their interfaces. A good picture of this complexity can be found in the work of M. J. Mutch², in which a complete band diagram of Cu/organosilica low-k interconnects is presented, illustrating the crucial role of interfaces in defining the leakage path.

In this context, the presence of unreacted metal oxide is not desirable since it may facilitate injection of charge carriers from the passivated Cu-lines into the conduction band of the MOF dielectric. To check for such effects, we performed IV-measurements on the 45 nm fork-fork capacitors without dielectrics (“air-gap”) and with gap-filling ZIF-8 and ZIF-67 layers (Supplementary Figure 13). The following conclusions can be drawn based on the recorded I-V curves:

- The pre-breakdown leakage current is comparable in all three devices. This observation can be attributed to the same nature of the low-field leakage current, likely associated with leakage through the SiCN/SiN_x passivation layer bridging both electrodes.
- The breakdown field is roughly two times lower in the devices containing gap-filling ZIFs as compared to the air-gap case. This observation means that the gap-filling medium provides additional leakage paths related to the unconverted metal oxide residues (ZnO_x, CoO_x) with a small band gap. This assumption is supported by the previously reported data on intrinsic leakage current in ZIF-8 films measured on simple metal-insulator-metal planar capacitor structures. In the absence of ZnO at the metal/insulator interface, the breakdown field was above 2 MV/cm¹.
- In the current experiment, both ZIF-8 and ZIF-67 demonstrate similarly low breakdown fields suggesting a similar failure mechanism involving interfacial metal oxide residues.

Although these I-V measurements do not allow to assess the intrinsic leakage through the ZIF-8 and ZIF-67 dielectrics, the adverse role of metal oxide residues

through degradation of electrical reliability was illustrated (even when the impact on interline capacitance is negligibly small as shown in section 3.4 of this document).

Supplementary Figure 13. Leakage current measured on 45 nm fork-fork capacitors featured with different gap-filling media. The ZIF-67 and ZIF-8 phases were formed via gas-phase conversion of oxidized CVD Co and ALD ZnO films, respectively.

1.3 Suggested references

Reviewer #1: *In this paper, the authors describe the integration of metal–organic frameworks (MOFs) as gap-filling low-k dielectrics in one-chip interconnects via a MOF-CVD process. The concept for using ZIF-8 as a low dielectric material was first presented by Eslava et al. in 2010. Since then, some significant studies on the low-k behavior of MOFs and their applications have been reported. e.g. Coord. Chem. Rev. 360, 71 (2018); Nat. Commun. 7, 11830 (2016); RSC Adv. 5, 45213 (2015) and J. Mater. Chem. C 2, 3762 (2014). With a fundamental study on dielectric properties and advancement in thin film growth techniques, MOFs were first proposed as future interlayer dielectric materials in 2015. e.g. ChemElectroChem 2, 786 (2015). The author is advised to cite all of the relevant literature as mentioned above in the “Introduction”.*

Authors’ response:

We would like to thank the reviewer for the suggested references. The proposed references have been appropriately cited in the introduction section of the manuscript.

Reviewer #2:

2.1 Clarification regarding the statements about zeolites made in the manuscript

Reviewer #2: Page 7: sentence “Another advantage of the ZIF-8 and ZIF-67 films in comparison with pure silica zeolites is that they are intrinsically hydrophobic due to the methyl group on the organic ligand pointing towards the pore interior“ is also arguable. Pure-silica zeolites are also intrinsically hydrophobic. It is either their composites with amorphous silica or poorly crystallized films that are not hydrophobic because of the hydroxyls presence in the amorphous part or on defects. Please rephrase the sentence to avoid confusion to readers. So far, highly crystalline pure-silica zeolite films despite being hydrophobic don't have enough porosity for a low k value. There are zeolite types with very high porosity that could show very low k value and excellent hydrophobicity, but their syntheses are too complicated, especially for films and with pure silica content. Please also correct “pure silica zeolite“ for “pure-silica zeolite“. “Pure” refers to “silica”, not to zeolites, and the papers the authors are citing are not strictly pure zeolites.

Authors' response:

We agree with the reviewer that the aforementioned sentence in the manuscript was somewhat misleading. The sentence was modified to eliminate ambiguity by highlighting the intrinsic hydrophobicity of pure-silica zeolites and the loss of hydrophobicity often encountered in the PSZ and PSZ-like thin films of these materials.

“Another advantage of the ZIF-8 and ZIF-67 frameworks is that they are intrinsically hydrophobic due to the methyl group on the organic ligand pointing towards the pore interior³. The MOF-CVD ZIF films do not absorb moisture, even at high relative humidity, as evidenced by water-based ellipsometric porosimetry. While pure-silica zeolite powders are also hydrophobic⁴, thin films of PSZ and PSZ-like materials often suffer from moisture uptake and related k-value increase due to silanol-rich grain boundaries or the presence of amorphous silica⁵⁻⁷.”

2.2 Challenges of MOFs integration into interconnects beyond gap-filling

Since similar concerns regarding the integration steps following the gap-filling step have been raised by the **Reviewer #3**, we decided to merge the feedback and address all the corresponding questions in this section.

Reviewer #2: After the CVD MOF grows between the Cu trenches, the films are in excess and kind of wavy. They will need to be planarized and cleaned for the formation of multilevel interconnects of more Cu trenches and insulator. How the authors envisage to continue the multilevel formation with such MOFs that are chemically and thermally not the most stable materials? A bit of discussion on this in the paper, acknowledging the challenges will ensure more rapid progress in the field and more citations.

Reviewer #3:

10) General question to give an outline to potential readers: How time-and cost intensive is the after-processing after the conversion step? As can be seen from the images of converted MOF on the fork structures, the MOF layer is highly inhomogenously grown.

11) How easily can the thickness be equalized and with which methods (etching will certainly not work)

12) How well suited are MOF layers as substrates for another gas phase deposition or other device fabrication related process. If the topology is like the '7 Blue Mountains with 7 dwarfs' then it will certainly be a problem!

Authors' response:

The polycrystalline nature of the MOF films makes them rougher than organosilica low-k coatings deposited by PECVD or spin-coating. Therefore, the first action to be taken for further integration of the MOF-CVD dielectric is indeed planarization. We propose two routes for the fabrication of MOF-based multi-layer interconnects, featuring two different planarization approaches. The challenges associated with each method are described below, followed by an outline of the aspects of MOF integration that would need more attention of the research community.

Supplementary Figure 15. Two metallization routes for a gap-filling MOF-CVD dielectric based on different planarization approaches and associated MOF integration challenges.

1) In *integration route I*, chemical-mechanical polishing (CMP) is used to remove the film roughness. CMP is a well-established method in microelectronics

technology used for the planarization of various material layers (e.g., SiO₂, W, Cu, Ru). It combines mechanical force (pressing a wafer against a rotating polishing pad) with chemical etching via a slurry (suspension of nanoparticles). The mechanical and chemical stability of MOFs will be key in determining the feasibility of this approach. The ZIFs studied in this work are promising in this respect as they withstand harsh conditions such as boiling alkaline water (NaOH) and organic solvents (benzene, methanol)⁸, though acidic slurry should be avoided due to their pH-sensitivity^{9,10}. Slurry-less polishing using only a polishing pad could be an alternative that has already been successfully applied to coatings with an ordered pore structure, namely periodic mesoporous silica¹¹. CMP of MOF films has not yet been explored. Finding a suitable approach that maintains the integrity and properties of the remaining MOF layer will be an essential research task to facilitate the integration of MOFs in microelectronics.

- 2) *Integration route II* uses a spin-on dielectric as a planarization coating. This extra dielectric could be a low-k variation of spin-on glass (SOG), which is routinely used in photolithography for local planarization before resist deposition. Because of the direct contact of the MOF-CVD layer with the SOG solution, possible compatibility issues should be considered, including chemical degradation of the MOF and irreversible adsorption of SOG components into the framework. These potential issues could be alleviated for alkaline SOG solutions containing relatively large silica/organosilica oligomers as used in the work of F. Goethals *et al.* for pore sealing applications¹².
- 3) Both proposed integration routes follow the same post-planarization steps and exploit the overburden layer formed during MOF-CVD as a low-k medium at the via (*i.e.* vertical contact) metallization level. This approach allows to reduce the complexity of the integration flow, making planarization the only MOF-specific step.

Nonetheless, there are several steps in the above flows that involve the MOF material for which compatibility needs to be further investigated: (step c) passivation with dielectric barrier; (step e) via hole dry etching; (step f) deposition of metal barrier. As in the case of MOF planarization, it is difficult to estimate the severity of the potential compatibility issues since the related research questions such as deposition-induced MOF damage, plasma-MOF interactions and MOF pore sealing remain currently unexplored. Still, some guidelines for minimizing the process-induced damage to the MOF layer can be gathered from a scarce number of relevant studies on MOFs and a vast number of reports on integration of porous organosilica low-k dielectrics.

The potential compatibility concerns for steps c and f are deposition-induced damage and infiltration of subsequent layers (or building blocks thereof) into the porous MOF layer. Deposition techniques commonly used for the deposition at these steps are PECVD, PVD, CVD and ALD. The first approach would not be preferred due to the high reactivity and diffusivity of the plasma species^{13,14}. Low-temperature (< 350°C) CVD and ALD methods could be an ideal choice if the in-diffusion of the precursor molecules into

MOFs can be suppressed¹⁵. Based on pore-sealing research on low-k organosilica films^{16,17}, it would be expected that MOFs with small pore apertures such as ZIF-8 / ZIF-67 (effective aperture size is approx. 4.0-4.2 Å)¹⁸ would be successfully sealed with ALD coating formed from relatively large precursor molecules exceeding the effective pore aperture.

When the interaction of MOF with plasma is unavoidable, such as for via opening (step e), ion-beam etching (ion milling) could be preferred over reactive ion etching due to its directionality and lack of reactive species which could diffuse into the pores of MOFs.

The above discussion has been partially included into the revised version of the manuscript.

2.3 Suggested grammar corrections

Reviewer #2:

- Some typos such as 2-methylimidazolate
- Please correct “cross-section SEM images” for “cross-sectional SEM images”
- Fig. 2b. If there are no numbers on y-axis, “a.u.” is not needed. “Intensity” would be more appropriate than “Count.”
- Fig. 2c. Please indicate which symbols are for adsorption and desorption

Authors' response:

We would like to thank the reviewer for highlighting spelling and grammar errors and for the suggested changes in the graphs. The proposed changes have all been incorporated into the revised version of the manuscript.

Reviewer #3:

3.1 Clarification regarding the experimentally observed expansion factor and non-idealities of metal precursor layers.

Reviewer #3:

2) p. 4 l.100ff MOF expansion by theory 19 - 17 times of the thickness of the oxide... based on theoretical crystal density estimation/calculation, but

2.1) how accurately are/is this model / estimation generally: How is the formation of defects and voids considered or taken into account? The XPS data (the only compositional proof) show imperfect stoichiometries. It may be the case that the model is not relevant for the rest of the MS, but after having a look at page 5 and the big discrepancies that they cannot properly explain, from the reviewer's perspective this becomes odd.

2.2) 5.8 nm ZnO — 24 nm ZIF-8 and 3.2 nm CoOx — 28.9 nm ZIF-67

3) Referring to page 5 and beginning of page 6: The model is obviously inaccurate - so how can the authors sell their approach as big breakthrough regarding the problems/issues connected to the prior described state of the art approach? They allegedly! (see later points) solve the problem with the void filling, but on page 6 they demonstrate themselves how difficult precise working with their method is: The expansion of the MOF is highly dependent on the nature of the ALD or native oxide layer and full conversion seemingly difficult to reach. At this state this method is in its infancy and certainly not suitable to be integrated in up-scaled device processing!

4) Also on p.6 they refer to results in another manuscript which they do not provide! As a reviewer we cannot check that - This not good practice and bold. They shall show it.

14) Line 142-148: The expanding factor is calculated based on bulk-crystalline materials "wurtzite hexagonal ZnO" and "cubic CoO", showing higher expansion values than the measured ones. Besides the mentioned explanations: How can the authors be sure that the calculated values are comparable, if the underlying oxide layers may be of amorphous nature? GI-XRD of CoO shows no reflexes and ZnO shows only very small and broad reflexes.

16) Line 148: "Non-ideality of the ALD ZnO film". What does non-ideality mean in this case?

Authors' response:

The XPS data is within the expected range for thin films of ZIF-8 and ZIF-67. Since the other comments above refer to the thickness expansion during the oxide-to-MOF conversion, we address the concerns raised by the reviewer together. The reference to the upcoming manuscript has been removed from the main text. The relevant data has been added to the Supplementary Info and is discussed below.

As was mentioned correctly by the reviewer, the expansion factor is mainly determined by the nature of the precursor. The theoretical expansion factor is calculated based on the metal density of ideal crystals of ZnO and CoO. This theoretical expansion factor is an upper limit since our precursor films contain defects and therefore have a lower metal content. The "effective" expansion factors

reported in the first version of the manuscript might have been misleading since they were calculated under the assumption of complete metal oxide conversion, which proved not correct. The following clarifications were added.

For the ALD ZnO film (45 DEZ/H₂O cycles at 120 °C), the volume concentration of Zn atoms was estimated by combining the Zn atom areal density ($30.8 \cdot 10^{15}$ at/cm²) assessed by Rutherford backscattering spectroscopy (Supplementary Figure 4) and the thickness (8.9 nm) measured by ellipsometry. The resulting density of Zn atoms equal to $3.45 \cdot 10^{22}$ at/cm³ is lower than that of wurtzite ZnO by approx. 18%, which results in the proportional reduction of the thickness expansion and an expansion factor of 13.9. The lower Zn content of the ALD ZnO film likely originates from pending Zn-OH groups that remain unreacted during the low-temperature ALD process¹⁹. Based on the corrected expansion factor, it can be calculated that roughly 4 nm ZnO remains at the interface between the ZIF-8 film and the substrate. However, due to its relatively high conductivity^{20,21}, the remaining thin ALD ZnO layer has negligible impact on the measured MIM capacitance.

Supplementary Figure 4. RBS spectrum (black) and fitting curve (red) recorded on 8.9 nm thick ALD ZnO film deposited on top of a Si/SiO₂ substrate.

The situation is more complicated for the native CoO_x layer formed on top of the sputtered Co film. While there is no doubt about the formation of ZIF-67 (based on GI-XRD, porosimetry and XPS data), the presence of metallic Co does not allow the same RBS methodology to correct the expansion factor and estimate how much CoO_x remains. Therefore, the theoretical expansion factor calculated for perfectly crystalline CoO was used (20.4) to estimate the maximum thickness of the remaining CoO_x layer as 1.8 nm. In reality, the remaining CoO_x layer will be thinner because of a lower effective expansion factor. Because of its higher resistivity, the presence of CoO_x at the interface between metallic Co and ZIF-67 should be taken into account for the estimation of the MOF k-value. However, the impact of this layer was negligible (section 3.3 of this document, Supplementary Figure 5).

3.2 Challenges of MOFs integration into interconnects beyond gap-filling

Reviewer #3:

10) General question to give an outline to potential readers: How time-and cost intensive is the after-processing after the conversion step? As can be seen from the images of converted MOF on the fork structures, the MOF layer is highly inhomogenously grown.

11) How easily can the thickness be equalized and with which methods (etching will certainly not work)

12) How well suited are MOF layers as substrates for another gas phase deposition or other device fabrication related process. If the topology is like the '7 Blue Mountains with 7 dwarfs' then it will certainly be a problem!

Authors' response:

Please, refer to section 2.2 in the feedback to Reviewer #2, where similar questions were addressed together.

3.3 Clarification regarding the k-value estimation (MIM capacitors)

Reviewer #3:

5) How accurate is ellipsometry for accessing the thickness of MOFs. Do MOFs absorb light of the ellipsometer spectrum? This is not clear... the authors made some thickness measurements for thicker films via TEM and SEM, but not for their devices they describe on page 7ff.... This reviewer is not too convinced when they write about the dielectric constants (k values) and don't explicitly say how thick the MOF films are, by the way. Maybe this reviewer overlooked it, but there seem to be no data on the film thickness for the MIM capacitors and the reviewer would like to know! as thickness influences the k value.

6) p. 7 lines 159-161 Why is it expected? Could the authors comment on it?

17) Line 158-161: "[...], a very thin CoOx layer (< 3 nm) could be still present underneath the MOF film. In both cases, the presence of thin interfacial oxide layers is expected to have negligible impact on the evaluation of dielectric and mechanical properties of the ZIF films, therefore we further treat them as parts of the corresponding MOF layers." If the presence of an interfacial oxide layer is supposed to have a negligible impact on dielectric properties, the authors are asked to cite a relevant paper that supports this idea. Even very thin interfacial oxides have an influence on the k-value determined in MIM-capacitor stacks. This was also the point made above regarding the thickness (point 5).

Authors' response:

The thickness of the ZIF-8 and ZIF-67 films used in the MIM capacitors has been determined based on SEM cross-sectional images recorded on the stacks with 70 nm Pt top electrodes (Supplementary Figure 3). As described in the "Methods" section of the manuscript, the extraction of average ZIF layer thickness (d) from the

obtained cross-sectional images was done with a custom Python script based on OpenCV (Open Source Computer Vision Library). This involved locating > 60 uniformly spaced points at the ZIF/Pt interface. These thickness values were also used to estimate the thickness of unconverted metal oxide (see section 3.1 of this document). Given the crucial role of the dielectric thickness, to accurately extract the k-value, we avoided using the thickness of the MOF films measured by ellipsometry. The accuracy of the latter is challenged by the stack complexity and associated modelling ambiguity.

As mentioned in section 3.1 of this document, both MIM stacks include a thin layer of unconverted metal oxide. In the case of the ZIF-8 capacitor, the influence of the underlying ZnO can be neglected due to its relatively high conductivity^{20,21}. The impact of the interfacial CoO_x on the extracted k-value was estimated by adding it as an extra capacitance in series with the ZIF-67 capacitance (Supplementary Figure 5). By assuming a k-value for interfacial CoO_x equal to that of CoO²², we found that even the worst-case scenario with the thickest and densest interfacial oxide layer results in minor changes of the extracted ZIF-67 k-value, not exceeding 0.05 and below the calculated error-bar (based on the film roughness).

Supplementary Figure 5. Impact of a residual CoO_x layer. (a) Dependence of the extracted ZIF-67 k-value on the thickness of the interfacial CoO_x layer; (b) equivalent circuit scheme for impedance measurements on the MIM stack.

3.4 Clarification regarding the effective k-value estimation (45 nm half-pitch fork-fork capacitors)

Reviewer #3:

7) p. 12 lines 292 - 294: Why does a higher k value imply that ZIF-67 layers are void free before TEM? And how should an electron beam create voids / holes randomly in a MOF layer.... It rather looks like imperfect growth especially in the SEM and TEM of the fork fork structures in the SI!!!! The concept works certainly, but not perfectly and they sell it as if it would – This is not acceptable.

8) p. 13 lines 295-298: This reviewer doesn't understand how the presence of rests of a conductive metal can increase the k of the insulating layer. They should explain their suspicion.

Authors' response:

As pointed out in the manuscript, the preparation of few tens of nanometers thick TEM lamella by ion milling and inspection under e-beam are rather damaging processes. In addition, MOFs are notoriously sensitive to beam damage. To demonstrate that the voids in the ZIF-67 phase observed by TEM (Supplementary Figure 8) are due to the beam damage or the preparation of the 30-50 nm thick lamellas, additional SEM cross-sectional images were recorded for the same fork-fork capacitors (Supplementary Figure 9). The cross-section surface was prepared by manually cleaving the substrate and depositing 1 nm of Pt to eliminate any beam-induced damage. No voids can be observed in these high-resolution SEM cross-sections.

Supplementary Figure 9. Cross-sectional SEM image of manually cleaved substrate with 45 nm half-pitch fork-fork capacitors.

CoO_x residues (formed as a result of metallic CVD Co oxidation evidenced by TEM EDS analysis in Supplementary Figure 7a) occur largely on top of the passivated Cu-lines and are caused by the non-uniform distribution of the CVD Co agglomerates with dimensions in some cases exceeding 10-15 nm. To estimate the possible impact of these defects on the extracted ZIF-67 k -value, we performed additional 2D capacitance simulations (Supplementary Figures 11-12). The k -value of the CoO_x residues was assumed to be equal to that of CoO ($k = 12.9$)²². Since these simulations are two-dimensional, defects defined in a cross-section geometry extend in the 3rd dimension, along the Cu-lines. Therefore, the effect of defects on the

simulated capacitance is strongly amplified and should be considered a worst-case scenario. The addition of unconverted CoO_x residues decreases the extracted ZIF-67 k-value by 0.35 for the largest simulated defect dimensions (Figure 4c).

Supplementary Figure 11. Modelled cross-sections of 45 nm half-pitch fork-fork capacitors featuring CoO_x residues of different size (defect radius R_{CoO_x}).

Supplementary Figure 12. The effect of CoO_x residues of different size on k-value of MOF-CVD ZIF-67 and on the total interline capacitance of a fork-fork device estimated by 2D capacitance simulations.

3.5 Remaining questions or remarks

Reviewer #3:

1) As the authors state themselves, the concept is not new but adopted for a proof of principle application (p 4 line 93ff)

Authors' response:

The current work is based on the method of gas-phase conversion of metal oxide films into MOFs ("MOF-CVD") pioneered by I. Stassen *et al.* The novelty of this work consists of the non-standard application of this method, namely as a tool to resolve the issue of narrow trench gap-filling with highly porous dielectric materials. The latter is particularly challenging for state-of-the-art porogen-based low-k dielectrics.

Reviewer #3:

9) Why are the ZIF-Layers in the SI figure 3 colorized? It looks as if they want to hide structural features! They shall decolorize it and just put colored lines on the interfaces between the layers.

Authors' response:

The original goal of the colorized SEM images was to visualize the employed method for estimation of the MOF layer thickness. The original SEM images have been added to the Supplementary Information document for comparison (Supplementary Figure 3a).

Reviewer #3:

13) Line 109: "Controlled oxidation of the metal surface" What are the specific parameters of the controlled oxidation? (Defined oxygen-containing atmosphere? Temperature?)

Authors' response:

In a regular thermal oxidation process, the Co oxidation rate as well as the type of formed CoO_x is largely defined by two factors: annealing temperature^{23,24} and partial oxygen pressure²⁵. The ideal implementation of the integration Route B described in the manuscript would be based on a cluster tool equipped with a vacuum transfer module and three process chambers for reduction of native CoO_x , controlled thermal oxidation of Co and for conversion of the freshly formed CoO_x into ZIF-67 (MOF-CVD process), respectively. However, in the current study, the CoO_x layer was formed by exposure of the Co layer to the cleanroom atmosphere for at least 48 h.

The appropriate references describing the Co oxidation process have been added to the revised manuscript.

Reviewer #3:

15) Figure 2: Mention that the given reference XRD pattern of ZIF-8 is almost the same as that of ZIF-67, as no reference pattern for ZIF-67 was shown.

Authors' response:

The simulated diffractogram for ZIF-67 has been added to Figure 2a. ZIF-8 and ZIF-67 are isostructural materials and have an almost identical diffraction pattern.

Reviewer #3:

18) Several references for the Figures are missing in the text (for example Figure 3b).

Authors' response:

The missing references to Figure 2d and Figure 3b have been added in the text of the revised manuscript.

References

1. Eslava, S. *et al.* Metal-Organic Framework ZIF-8 Films As Low- κ Dielectrics in Microelectronics. *Chem. Mater.* **25**, 27–33 (2013).
2. Mutch, M. J. *et al.* Band diagram for low- κ /Cu interconnects: The starting point for understanding back-end-of-line (BEOL) electrical reliability. *Microelectron. Reliab.* **63**, 201–213 (2016).
3. Ortiz, A. U., Freitas, A. P., Boutin, A., Fuchs, A. H. & Coudert, F. X. What makes zeolitic imidazolate frameworks hydrophobic or hydrophilic? the impact of geometry and functionalization on water adsorption. *Phys. Chem. Chem. Phys.* **16**, 9940–9949 (2014).
4. Flanigen, E. M. *et al.* Silicalite, a new hydrophobic crystalline silica molecular sieve. *Nature* **271**, 512–516 (1978).
5. Li, Z., Li, S., Luo, H. & Yan, Y. Effects of crystallinity in spin-on pure-silica-zeolite MFI low-dielectric-constant films. *Adv. Funct. Mater.* **14**, 1019–1024 (2004).
6. Eslava, S. *et al.* Ultraviolet-assisted curing of polycrystalline pure-silica zeolites: Hydrophobization, functionalization, and cross-linking of grains. *J. Am. Chem. Soc.* **129**, 9288–9289 (2007).
7. Eslava, S. *et al.* Characterization of spin-on zeolite films prepared from Silicalite-I nanoparticle suspensions. *Microporous Mesoporous Mater.* **118**, 458–466 (2009).
8. Park, K. S. *et al.* Exceptional chemical and thermal stability of zeolitic imidazolate frameworks. *Proc. Natl. Acad. Sci.* **103**, 10186–10191 (2006).

9. Sun, C.-Y. *et al.* Zeolitic imidazolate framework-8 as efficient pH-sensitive drug delivery vehicle. *Dalt. Trans.* **41**, 6906 (2012).
10. Wu, C., Xiong, Z., Li, C. & Zhang, J. Zeolitic imidazolate metal organic framework ZIF-8 with ultra-high adsorption capacity bound tetracycline in aqueous solution. *RSC Adv.* **5**, 82127–82137 (2015).
11. Arnold, D. C. *et al.* Planarized and Nanopatterned Mesoporous Silica Thin Films by Chemical-Mechanical Polishing of Gap-Filled Topographically Patterned Substrates. *IEEE Trans. Nanotechnol.* **10**, 451–461 (2011).
12. Goethals, F. *et al.* A new procedure to seal the pores of mesoporous low-k films with precondensed organosilica oligomers. *Chem. Commun.* **48**, 2797–2799 (2012).
13. Decoste, J. B., Peterson, G. W., Smith, M. W., Stone, C. A. & Willis, C. R. Enhanced Stability of Cu-BTC MOF via Perfluorohexane Plasma-Enhanced Chemical Vapor Deposition. *J. Am. Chem. Soc.* **134**, 1486–1489 (2012).
14. DeCoste, J. B., Rossin, J. A. & Peterson, G. W. Hierarchical Pore Development by Plasma Etching of Zr-Based Metal-Organic Frameworks. *Chem. - A Eur. J.* **21**, 18029–18032 (2015).
15. Lemaire, P. C., Lee, D. T., Zhao, J. & Parsons, G. N. Reversible Low-Temperature Metal Node Distortion during Atomic Layer Deposition of Al₂O₃ and TiO₂ on UiO-66-NH₂ Metal-Organic Framework Crystal Surfaces. *ACS Appl. Mater. Interfaces* **9**, 22042–22054 (2017).
16. Jourdan, N. *et al.* Study of Chemical Vapor Deposition of Manganese on Porous SiCOH Low-k Dielectrics Using Bis(ethylcyclopentadienyl)manganese. *Electrochem. Solid-State Lett.* **15**, H176 (2012).
17. Sun, Y. *et al.* Impact of Plasma Pretreatment and Pore Size on the Sealing of Ultra-Low- k Dielectrics by Self-Assembled Monolayers. *Langmuir* **30**, 3832–3844 (2014).
18. Krokidas, P., Moncho, S., Brothers, E. N., Castier, M. & Economou, I. G. Tailoring the gas separation efficiency of metal organic framework ZIF-8 through metal substitution: a computational study. *Phys. Chem. Chem. Phys.* **20**, 4879–4892 (2018).
19. Beh, H. *et al.* Quasi-metallic behavior of ZnO grown by atomic layer deposition: The role of hydrogen. *J. Appl. Phys.* **122**, 025306 (2017).
20. Min, Y.-S., An, C.-J., Kim, S.-K., Song, J.-W. & Hwang, C.-S. Growth and Characterization of Conducting ZnO Thin Films by Atomic Layer Deposition. *Bull. Korean Chem. Soc.* **31**, 2503–2508 (2010).
21. Beh, H., Hiller, D., Laube, J., Gutsch, S. & Zacharias, M. Deposition temperature dependence and long-term stability of the conductivity of undoped ZnO grown by atomic layer deposition. *J. Vac. Sci. Technol. A Vacuum, Surfaces, Film.* **35**, 01B127 (2017).
22. Rao, K. V. & Smakula, A. Dielectric Properties of Cobalt Oxide, Nickel Oxide, and Their Mixed Crystals. *J. Appl. Phys.* **36**, 2031–2038 (1965).
23. Birks, N., Meier, G. H. & Pettit, F. S. Oxidation of pure metals. in *Introduction to the High-Temperature Oxidation of Metals* 75–100 (Cambridge University Press). doi:10.1017/CBO9781139163903.006
24. Tompkins, H. G. & Augis, J. A. The oxidation of cobalt in air from room temperature to 467°C. *Oxid. Met.* **16**, 355–369 (1981).
25. Bridges, D. W., Baur, J. P. & Fassell, W. M. Effect of Oxygen Pressure on the Oxidation Rate of Cobalt. *J. Electrochem. Soc.* **103**, 614 (1956).

REVIEWERS' COMMENTS:

Reviewer #1 (Remarks to the Author):

The authors addressed most comments provided by all reviewers. Publication of this manuscript in Nature Communications is recommended.

Reviewer #2 (Remarks to the Author):

The authors have addressed all the comments of the reviewers and its new version is ready for publication without changes.

Regards,

Dr Salvador Eslava

University of Bath